# Implications of Pleural Fluid Composition in Persistent Pleural Effusion following Orthotopic Liver Transplant

**DOI:** 10.3390/medsci11010024

**Published:** 2023-03-17

**Authors:** Bhavesh H. Patel, Kathryn H. Melamed, Holly Wilhalme, Gwenyth L. Day, Tisha Wang, Joseph DiNorcia, Douglas Farmer, Vatche Agopian, Fady Kaldas, Igor Barjaktarevic

**Affiliations:** 1David Geffen School of Medicine at UCLA, Los Angeles, CA 90095, USA; bhaveshpatel@mednet.ucla.edu; 2Division of Pulmonary and Critical Care Medicine, Department of Medicine, David Geffen School of Medicine at UCLA, Los Angeles, CA 90095, USA; kmelamed@mednet.ucla.edu (K.H.M.);; 3Department of Medicine Statistics Core, David Geffen School of Medicine at UCLA, Los Angeles, CA 90095, USA; 4Department of Medicine, David Geffen School of Medicine at UCLA, Los Angeles, CA 90095, USA; 5Division of Liver and Pancreas Transplantation, Department of Surgery, David Geffen School of Medicine at UCLA, Los Angeles, CA 90095, USA

**Keywords:** hepatic hydrothorax, orthotopic liver transplantation, exudative pleural effusion, Light’s criteria

## Abstract

Persistent pleural effusions (PPEf) represent a known complication of orthotopic liver transplant (OLT). However, their clinical relevance is not well described. We evaluated the clinical, biochemical, and cellular characteristics of post-OLT PPEf and assessed their relationship with longitudinal outcomes. We performed a retrospective cohort study of OLT recipients between 2006 and 2015. Included patients had post-OLT PPEf, defined by effusion persisting >30 days after OLT and available pleural fluid analysis. PPEf were classified as transudates or exudates (Exud^Light^) by Light’s criteria. Exudates were subclassified as those with elevated lactate dehydrogenase (Exud^LDH^) or elevated protein (Exud^Prot^). Cellular composition was classified as neutrophil- or lymphocyte-predominant. Of 1602 OLT patients, 124 (7.7%) had PPEf, of which 90.2% were Exud^Light^. Compared to all OLT recipients, PPEf patients had lower two-year survival (HR 1.63; *p* = 0.002). Among PPEf patients, one-year mortality was associated with pleural fluid RBC count (*p* = 0.03). While Exud^Light^ and Exud^Prot^ showed no association with outcomes, Exud^LDH^ were associated with increased ventilator dependence (*p* = 0.03) and postoperative length of stay (*p* = 0.03). Neutrophil-predominant effusions were associated with increased postoperative ventilator dependence (*p* = 0.03), vasopressor dependence (*p* = 0.02), and surgical pleural intervention (*p* = 0.02). In summary, post-OLT PPEf were associated with increased mortality. Ninety percent of these effusions were exudates by Light’s criteria. Defining exudates using LDH only and incorporating cellular analysis, including neutrophils and RBCs, was useful in predicting morbidity.

## 1. Introduction

Pleural effusion is a common complication of orthotopic liver transplant (OLT), reported to occur in 32% to 95% of cases within the first week after transplant [1,2,3,4,5,6,7,8]. The majority of these effusions are thought to be caused by lymphatic disruption and diaphragmatic defects occurring during hepatectomy, leading to the transfer of fluid from ascites into the pleural space [9,10]. Accordingly, these transient post-transplant effusions are predominately located in the right pleural space near the site of surgical manipulation, often with a transudative composition, similar to that of cirrhotic ascites [7,11,12]. Notably, most of these early postoperative pleural effusions are transient and often clinically insignificant, and do not need drainage by thoracentesis or tube thoracostomy [11,13].

However, up to one-quarter of these effusions persist greater than one week, of which nearly two-thirds require pleural intervention [13]. Among persistent pleural effusions, exudative effusions can occur with some regularity and up to one-fifth have micro-organisms identified on culture [1]. In contrast to transient pleural effusions, persistent effusions have been reported to portend worse outcomes, associated with atelectasis, pneumonia, and empyema [1,5,13]. Recurrent pleural effusions have also been associated with allograft rejection [11]. However, these small-scale studies have not clearly identified specific risk factors for poor outcomes within this population nor have they clearly described the pathophysiologic mechanisms driving persistent effusion.

While pleural fluid (PF) analysis is often performed when a persistent effusion is found, the diagnostic and prognostic value of PF biochemistries and cellular composition in this clinical context is unclear. It is reported that a majority of transient post-transplant pleural effusions are transudates [7,11,12], but the composition of persistent effusions requiring pleural intervention is more diverse and has not been well studied. Furthermore, when these effusions have been described, they are dichotomized using the traditional criteria for transudates and exudates as described by Light in 1972 [14]. However, this classification lacks specificity and may not have clinical utility in this postoperative cohort. Some data suggest that distinguishing exudative pleural effusions after OLT specifically by protein or by lactate dehydrogenase (LDH) may have more clinical utility [15]. Lastly, the clinical relevance of PF neutrophilic inflammation, specifically spontaneous bacterial empyema (defined by neutrophil count >500 cells/μL or positive culture with neutrophil count >250 cells/μL in the absence of a parapneumonic effusion), has been well described in cirrhotic patients with hepatic hydrothorax [16,17]. In contrast, the significance of PF neutrophils and cellular composition in post-OLT persistent effusions has not been clearly described.

In this retrospective analysis, we hypothesized that the traditional definition of exudative pleural effusion by Light’s criteria is poorly associated with clinical outcomes, and aimed to search for specific clinical, biochemical, and cellular biomarkers predictive of adverse outcomes in OLT recipients with persistent pleural effusions. To our knowledge, no study has investigated PF composition in such a way with a large sample size of persistent effusions after OLT. 

## 2. Materials and Methods

### 2.1. Design, Setting, and Population

We performed a retrospective cohort study of all patients between January 2006 and October 2015 who received an orthotopic liver transplant at our high-volume, quaternary liver transplant center. Patients were included as part of the persistent pleural effusion (PPEf) cohort if they were 18 years or older at the time of transplantation, had a persistent pleural effusion, and had pleural fluid sampling with pleural fluid analysis within the first post-transplant year. Persistent effusion was defined as a pleural effusion that was radiographically present both immediately postoperatively and at a timepoint at least 30 days after OLT. Our use of 30 days to define persistence mirrors other authors’ definition of persistent ascites after OLT and persistent pleural effusion after cardiac surgery [18,19,20,21]. This study was approved by the institution’s internal review board (IRB protocol #14-000365).

### 2.2. Data Collection

For all adult OLT patients within the study timeframe, we collected age, sex, physiologic Model for End-Stage Liver Disease (MELD) score prior to transplant, and survival data. MELD score was calculated based on serum bilirubin, INR, and serum creatinine. Serum sodium was not included since this lab value was introduced into the MELD calculation in 2016, after the study timeframe. Since mortality risk has been shown to correlate with MELD scores even above the listing capped score of 40, we chose not to cap MELD scores to better capture patients’ disease severity for this analysis [22]. 

For patients who met our inclusion criteria for the PPEf cohort, we also recorded additional variables. We collected preoperative characteristics, including etiology of liver failure and comorbid conditions such that the Charlson comorbidity index (CCI) could be calculated [23]. Post-OLT data collected included peak total bilirubin, peak alanine transaminase, and peak aspartate transaminase between day 8 and 1 year after transplant. Additionally, PF data were recorded for the most proximate sampling after OLT within the first post-transplant year, and included cell count and differential, microbiology data, LDH, and protein concentration. PF composition was classified as exudate by Light’s criteria (Exud^Light^), which required any one of the following be met: PF to serum protein ratio greater than 0.5, PF to serum LDH ratio greater than 0.6, or PF LDH greater than two-thirds of the upper limit of normal serum LDH. PPEf that did not meet any of these criteria were classified as transudates. Finally, for the purposes of correlating with clinical outcomes, we subclassified Exud^Light^ into Exud^LDH^ and Exud^Prot^, which were two non-mutually exclusive groups. Exud^LDH^ were defined as any exudate which met the LDH criteria (irrespective of the protein criterion), and Exud^Prot^ was defined as any exudate which met the protein criterion (irrespective of the LDH criteria). Any PPEf that did not meet the criteria for Exud^LDH^ or Exud^Prot^ was classified as a non-exudate, specifically Non-Exud^LDH^ and Non-Exud^Prot^, respectively. Non-Exud^LDH^ included traditional transudates and exudates defined only by protein. Non-Exud^Prot^ included traditional transudates and exudates defined only by LDH. Regarding PF cellular composition, neutrophil predominance (Ne-predominance) or lymphocyte predominance (Lym-predominance) was defined as neutrophils or lymphocytes composing greater than 50% of the total PF leukocytes respectively. Spontaneous bacterial empyema (SBEM) was defined as PF neutrophil count >500 cells/μL or positive culture with PF neutrophil count >250 cells/μL, in the absence of a parapneumonic effusion. These criteria have classically been used to define spontaneous infection of hepatic hydrothorax in cirrhosis, often seen with accompanying spontaneous bacterial peritonitis [16,17]. Here, we apply this classification to our patients who are recently post-OLT and subject to some of the same shifting hydrostatic forces that drive ascites and pleural effusion development in cirrhosis.

Pleural interventions were defined as thoracentesis, tube thoracostomy, video-assisted thoracoscopic surgery pleurodesis, and tunneled pleural catheter insertion. Surgical pleural intervention was defined as all the aforementioned interventions except thoracentesis. Clinical endpoints included one-year and two-year survival, total hospital days within the first post-transplant year, postoperative length of stay (LOS), pleural space interventions, need for hemodialysis, mechanical ventilation, and vasopressor requirement greater than 14 days after OLT. 

### 2.3. Data Analysis

As a primary analysis, one- and two-year survival for all OLT patients vs. PPEf patients was analyzed using Cox proportional hazards regression with effect sizes reported hazard ratios, adjusting for age and MELD score. As a sensitivity analysis, the same comparison was also analyzed with the restricted mean time lost (RMTL), which performs similarly to proportional hazards modeling but can approximate the hazard ratio when the assumption of proportional hazards is not satisfied [24,25]. Within the PPEf cohort, pre- and postoperative clinical characteristics were described for all patients as well as compared between those alive and deceased at one year using Student’s *t*-test for continuous variables and Fisher’s exact test for categorical variables. All continuous variables are reported as mean ± standard deviation, except LOS which is reported as median (interquartile range). Outcomes were then compared for the three classifications of exudate defined in this study (Exud^Light^, Exud^Prot^, and Exud^LDH^) versus transudates or non-exudates. Similarly, outcomes were compared for PPEf with Ne-predominance or Lym-predominance versus those without the respective predominance. One-year survival was analyzed using Cox proportional hazards regression (effect size reported as hazard ratio). Dichotomous outcomes were analyzed using logistic regression (effect size reported as odds ratio). For comparisons in which there were zero events in one of the groups, the Firth bias correction for logistic regression was used to generate odds ratios. Finally, postoperative LOS was analyzed using Student’s *t*-test after log-normalizing (effect size reported as mean difference). All outcome models were adjusted for age and MELD score. All statistical tests were two-sided with *p* < 0.05 defined as statistical significance. Analyses were conducted using Stata (StataCorp LP, College Station, TX, USA) and SAS (SAS Institute, Cary, NC, USA).

## 3. Results

### 3.1. Clinical Characteristics of OLT Recipients with PPEf

Of 1602 total OLT patients identified during the study timeframe, 124 (7.7%) met inclusion criteria for PPEf during the first post-transplant year. There were no significant differences in age, sex, or MELD score between the overall cohort of OLT patients and PPEf patients (Table A1). Those with PPEf had similar 1-year survival (79.4% vs. 84.5%) with HR 1.13 (95% CI 0.75–1.69, *p* = 0.56), but lower 2-year survival (63.5% vs. 79.1%) with HR 1.63 (95% CI 1.19–2.22, *p* = 0.002) compared to the overall OLT cohort, adjusting for age and MELD score with Cox proportional hazards regression (Figure 1). These results were consistent with RMTL analysis of the survival data: those with PPEf had similar 1-year RMTL (33.5 vs. 37.9 days) with a RMTL ratio of 0.88 (95% CI 0.58–1.35; *p* = 0.57), but higher 2-year RMTL (139.8 vs. 103.9 days) with a RMTL ratio of 1.35 (95% CI 1.00–1.80; *p* = 0.047).

As presented in Table 1, the majority of the PPEf patients had an infectious etiology of liver disease (hepatitis B and C), with a high prevalence of hepatocellular carcinoma (37.1%). Pre-existing chronic pulmonary disease, including COPD (4.0%) and asthma (5.6%), was infrequent, while comorbidities related to liver disease, including hepatopulmonary syndrome (23.4%) and hepatorenal syndrome (57.3%), were common. While only 3.2% of PPEf patients required surgical interventions prior to OLT, 42.7% required such intervention in the first postoperative year. Kidney injury requiring hemodialysis after OLT was frequent (42.7%). Of the 14 positive PF cultures after OLT, most were bacterial and one was mycobacterial, and none were fungal infections. The median LOS for the index OLT hospitalization was 33 (19–75.5) days. The majority of patients (54.0%) had a postoperative LOS greater than 30 days. Of the 46.0% patients who were discharged prior to 30 days, all required at least one readmission during the first post-transplant year (mean 3.5 ± 1.8 readmissions). And, of the patients discharged before 30 days, the median total hospital days within the first postoperative year was 54 (34–72) days.

### 3.2. Predictors of 1-Year Mortality in PPEf Patients

To identify the predictors of outcomes among OLT recipients with PPEf, this cohort was divided based on 1-year survival (Table 1). Twenty-seven PPEf patients (21.8%) died within the first year. Those who died were older (58.3 ± 5.5 vs. 54.1 ± 10.0 years, *p* = 0.04), had higher CCI (6.8 ± 1.7 vs. 5.6 ± 1.7, *p* = 0.002), and a higher rate of pre-existing chronic kidney disease (25.9% vs. 8.2%, *p* = 0.02). Postoperatively, patients who died had higher peak total bilirubin (15.5 ± 13.0 vs. 8.2 ± 8.0 mg/dL, *p* < 0.001) and longer total LOS in the first postoperative year (81 (64.5–156) vs. 72 (47–117) days, *p* = 0.049) compared to survivors. We did not find mortality to be associated with prolonged mechanical ventilation, vasopressor use, hemodialysis dependence, need for surgical pleural intervention, or positive PF culture.

### 3.3. Clinical Significance of Exudates Defined by Light’s Criteria

Of the 124 PPEf patients with PF analysis, 123 had PF biochemistry data available. Of these, 111 (90.2%) were exudates (Exud^Light^) and only 12 were transudates by Light’s criteria. A majority (62.0%) of post-transplant effusions were also present prior to OLT, and of those with available PF biochemistries pre-transplant (N = 19), 53% were exudative.

Table 2 summarizes baseline characteristics and clinical outcomes for post-OLT Exud^Light^ vs. transudates. While no significant difference was found in age or CCI, transudates had higher transplant MELD score (34.6 ± 8.7 vs. 27.1 ± 12.7, *p* = 0.049). Adjusted for age and MELD, there were no significant differences in outcomes, including 1-year survival, although Exud^Light^ trended towards longer LOS.

### 3.4. Clinical Significance of Protein—vs. LDH-Based Definition of Exudates 

Of the 123 PPEf patients with PF biochemistry data, 91 patients had both PF protein and LDH data available, allowing for the subclassification of Exud^Light^ into Exud^LDH^ and Exud^Prot^. There were no significant differences in age, sex, MELD score, or CCI between the overall cohort of PPEf patients and the 91 patients included in this subanalysis (Table A2). A majority of these effusions were exudates by Light’s criteria (83/91, 91.3%), of which 63 were Exud^LDH^ and 75 were Exud^Prot^. Table 3 summarizes outcomes for Exud^LDH^ and Exud^Prot^ compared to the effusions that did not meet these respective classifications (Non-Exud^LDH^ or Non-Exud^Prot^). Neither the presence of Exud^LDH^ nor Exud^Prot^ was associated with 1-year survival. However, Exud^LDH^ were associated with increased postoperative ventilator dependence (30.2% vs. 7.1%, OR 5.59, *p* = 0.03) and postoperative LOS (48 (20.5–85) vs. 29 (15–49) days, *p* = 0.03), when adjusted for age and MELD score. Conversely, Exud^Prot^ were not associated with any clinical outcomes.

### 3.5. Significance of PF Cellular Content

PF cell count differentials were available for 112 patients. Exud^LDH^ were associated with more pleural fluid neutrophilia (30% ± 32% vs. 15% ± 21%, *p* = 0.04) as well as a higher incidence of SBEM (17.9% vs. 0%, *p* = 0.03) compared to Non-Exud^LDH^. When classifying by Light’s criteria (Exud^Light^) or by protein (Exud^Prot^), exudates had no significant differences in cell counts or association with SBEM when compared to non-exudates.

While total PF WBC count was not associated with any difference in outcomes, Table 4 depicts the outcomes of PPEf classified by Ne-predominance (26/112, 23.2%) or Lym-predominance (33/112, 29.5%). Ne-predominant effusions were associated with increased postoperative ventilator dependence (34.6% vs. 15.1%, *p* = 0.03), vasopressor dependence (15.4% vs. 4.7%, *p* = 0.02), and surgical pleural intervention (57.7% vs. 32.6%, *p* = 0.02). Conversely, Lym-predominant effusions were associated with decreased ventilator dependence (6.1% vs. 25.3%, *p* = 0.03).

Lastly, increased PF RBC count was associated with one-year mortality (165 × 10^3^/μL ± 425 × 10^3^/μL in those deceased vs. 52.5 × 10^3^/μL ± 133 × 10^3^/μL in those alive, *p* = 0.03). Additionally, Exud^LDH^ had a higher incidence of effusions with PF RBC count greater than 100 × 10^3^/μL compared to Non-Exud^LDH^, although this was not statistically significant (11/59, 18.6% vs. 1/28, 3.6%; *p* = 0.09). Of the 16 PPEf patients who had PF RBC counts greater than 100 × 10^3^/μL, 6 (38%) had died at one year and 8 (50%) required surgical pleural intervention. In addition, PF RBC counts were significantly higher in patients with HCC than those without HCC (137 × 10^3^/μL ± 353 × 10^3^/μL vs. 43.3 × 10^3^/μL ± 96.2 × 10^3^/μL, *p* = 0.03). This relationship did not hold true for any other etiology of pre-OLT liver disease.

## 4. Discussion

We found that pleural effusions, persisting at least 1 month after OLT and clinically significant enough to require pleural sampling, were present in 7.7% of OLT recipients at our institution. These PPEf patients had significantly decreased two-year survival compared to all OLT patients when adjusted for age and MELD score. Additionally, elevated PF LDH, neutrophils, and RBCs were associated with worse clinical outcomes. To our knowledge, this is the largest study that shows an association between persistent post-OLT effusion and decreased survival.

The clinical presentation of post-OLT pleural effusion is highly variable. These effusions are often transient, asymptomatic, and mostly transudative in composition [7,12,13], and often managed by medical management and observation [9,26]. However, certain effusions persist and require invasive intervention [27]. Prior data have shown that post-transplant pleural effusions are associated with postoperative morbidity, but not mortality [5,27]. Smaller studies have shown that the subset of patients with effusions requiring thoracentesis and tube thoracostomy have lower survival [5,27]. 

Since the majority of effusions after OLT are transient, we focused on those that persist, and defined them by their presence at least 30 days post-OLT. This definition mirrors that of persistent ascites after OLT and persistent pleural effusion after cardiac surgery [18,19,20,21]. By examining patients with only persistent effusion, we selected for a sicker subset of patients, resulting in a higher MELD score and longer LOS than that which is reported in the literature for the average OLT cohort [28]. Furthermore, we limited our analysis to only effusions with available PF analysis data, which offers two major benefits: each PF sampling was performed per usual care, thus satisfying the criterion for clinical relevance, and having PF biochemical and cellular data allowed for a better understanding of the etiology and clinical relevance of these effusions. We sought to find an association between these PF characteristics and both long-term (survival, LOS, surgical pleural intervention) and short-term (immediate postoperative hemodialysis, ventilator dependence, vasopressor requirement) outcomes. We chose to include these three shorter-term outcomes as we believe they are valuable markers of post-transplant morbidity and represent an at-risk, sicker cohort of patients with systemic disease.

While data show that early post-transplant pleural effusions are often transudates [7,12,13], there is a paucity of literature on the biochemical composition of persistent pleural effusions. We only found one such study by Bozbas et al., which reported that among post-OLT effusions requiring thoracentesis (N = 37), 43.2% were exudative [1]. While the composition of pre-OLT effusions in our cohort was balanced between exudates and transudates (53% and 47%, respectively), an overwhelming majority (90.2%) of our post-OLT PPEfs were exudative by Light’s criteria. The discrepancy between our results and those of Bozbas et al. may be explained by the fact that their cohort was much smaller and included early postoperative effusions, thus offering less generalizability in the context of persistent effusions after OLT. Bozbas et al. also found that 18.9% had a positive PF culture, which was associated with increased mortality within 90 days post-transplant. This is similar to our 11.3% positive PF culture incidence, but we did not find an association with 1-year mortality, perhaps because pleural infection has a greater impact on early rather than late post-transplant mortality.

The etiologies of exudates after liver transplantation are varied and complex. They may include parapneumonic, trauma from pleural space manipulation, malignancy, and subdiaphragmatic inflammation including pancreatic and biliary disease [29]. To further delineate the etiology of and risk factors for persistent effusions, we examined the PF composition in more detail. We focused our biochemical analysis on PF LDH and protein as sensitive markers of exudative vs. transudative etiology [14], rather than other PF features. There was no association between clinical outcomes and patients who had exudative effusions defined by Light’s criteria. However, when defining exudates by LDH only, thereby excluding exudates which met criteria by only pleural fluid protein, we found correlation with worse outcomes including postoperative ventilator dependence and LOS. Furthermore, patients with LDH-defined exudates had increased pleural neutrophilic inflammation as well as SBEM, which has been associated with increased mortality in cirrhotic patients in the prior literature and increased morbidity in our cohort [16,17]. On a biochemical level, LDH is a ubiquitous cellular enzyme and a sensitive marker of cell death [30]. Thus, LDH-rich fluid may suggest an inflammatory pleural effusion etiology driven by an active process rather than a simple third-spacing of fluid into body cavities [31,32]. Chronic inflammatory pleural effusions have been associated with worse outcomes in OLT patients, and can even lead to the development of trapped lung [33]. Our findings add to the current understanding of pleural effusion biomarkers and their ability to predict outcomes in OLT recipients, and are aligned with previously published data suggesting that elevated PF LDH is an indicator of active pleural inflammation while protein is a sign of abnormal pleural lymphatics and capillary permeability [15,29]. We show that the LDH component of Light’s criteria may be a more sensitive marker of poor outcomes than the protein component in this cohort.

Neutrophils have also been shown to be markers of acute pleural inflammation, [34] and PF LDH >1000 and Ne-predominance are both associated with complex parapneumonic effusion and empyema [29,34,35]. Not surprisingly, our data confirm the clinical relevance of neutrophil-predominant PF composition and show its association with longer postoperative ventilator time, vasopressor dependence, and increased surgical pleural intervention. Interestingly, the presence of RBCs in the fluid was associated with increased risk of mortality, and there was a trend towards a positive association between elevated RBCs (defined as greater than 100 × 10^3^/μL) and Exud^LDH^. PF RBC counts above this level are strongly suggestive of trauma, malignancy, or pulmonary infarction [35]. We cannot be certain of the exact cause of the PF RBC elevations in our cohort. However, in the context of post-OLT PPEf patients, one can speculate that iatrogenic trauma during invasive pleural intervention may be implicated. In our cohort, 50% of PPEf with elevated PF RBC required a chest tube, video-assisted thoracoscopic surgery, or tunneled pleural catheter after transplant.

Taken together, PF LDH, neutrophils, and RBC count have prognostic value in post-OLT PPEf patients, showing associations with both long- and short-term outcomes. These short-term outcomes, namely postoperative ventilator dependence and vasopressor requirement, occurred prior to the determination of a persistent effusion. This temporal relationship implies that these effusions may not be drivers of these outcomes, but their presence is, nonetheless, associated with systemic disease. Therefore, our data suggest that these effusions often are the result of complex, exudative fluid with pleural inflammation, perhaps due to local infection, trauma, or severe systemic disease. We posit that these persistent effusions are an indicator of ongoing systemic inflammatory processes that themselves can increase risk for poor outcomes in post-transplant patients. As such, persistent pleural effusion can serve as an indicator of an at-risk patient, and we recommend that close and judicious care be paid to this cohort.

Our study has certain limitations. First, data were obtained retrospectively in a single center with a relatively small sample size with instances of incomplete data. Specifically, the comparison between Exud^Light^ and transudates (Table 2) was restricted by a small comparison group with transudates (N = 12). Larger studies are needed to make definitive conclusions on the utility of Light’s criteria in this comparison. Therefore, with our data, we are not able to conclude that Light’s criteria do not have any prognostic value in this cohort. However, our results suggest that PF LDH, neutrophils, and RBCs may hold additional diagnostic and prognostic value, and they can be used in conjunction with, not in place of, Light’s criteria. This is supported by the data that show that, even with our small sample size, PF LDH was associated with indicators of increased morbidity, whereas PF protein was not. Second, our primary analysis of survival between all OLT patients and PPEf patients (Figure 1) indicates that our data do not satisfy the proportional hazards assumption since the survival curves cross each other. PPEf patients appear to have better survival early on for a short period of time, then convert to having worse survival within the first year. This may, in part, be driven by our inclusion criteria for PPEf who, by definition, must survive at least 1 month or longer to have a persistent effusion. To address the non-proportionality, we included a sensitivity analysis with RMTL, which was consistent with the primary analysis. This supports the finding that there was a statistically significant difference in two-year survival between groups, regardless of the proportional hazards assumption. Third, our study is also limited by its observational design, which only included patients who had PF sampling as decided by the managing clinician. This may select for sicker patients and exclude others with milder disease. Fourth, since PF sampling was an inclusion criterion, our study does not capture unsampled persistent effusion and thus may not be representative of all patients with clinically significant pleural effusions. Finally, we could not control the timing of PF sampling. We selected the most proximate sampling after OLT, which represents only a snapshot of PF composition. This is important since the timing of PF analysis in relation to acute pleural injury is known to influence the cellular population and biochemical features.

Nevertheless, our study has strengths worth emphasizing. To our knowledge, this is the largest study to date to report on the clinical significance of the biochemical and cellular composition of persistent pleural effusions after OLT. This is one step towards advancing our understanding of the pathophysiology driving these persistent effusions and better identifying a group of patients at high risk for post-transplant morbidity and mortality.

## 5. Conclusions

Persistent pleural effusion after liver transplantation is not uncommon in OLT recipients and is associated with increased mortality. Within our PPEf cohort, 90% of pleural effusions were exudates by the Light’s criteria. Defining exudates using LDH levels and incorporating the cellular analysis, including both neutrophils and RBCs, was useful in predicting post-transplant morbidity. Our study adds value by revealing several important prognostic factors that clinicians can use in conjunction with Light’s criteria in managing persistent postoperative effusions. Future work should focus on prospective interventions for this at-risk post-transplant cohort.

## Figures and Tables

**Figure 1 medsci-11-00024-f001:**
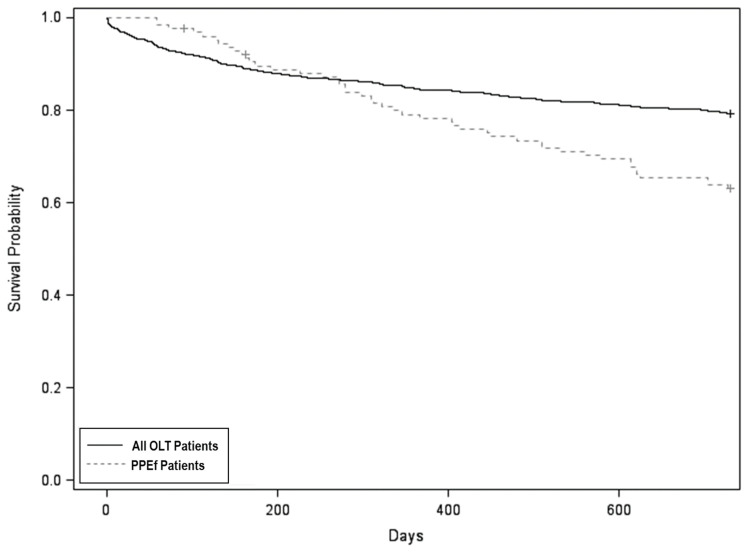
Survival probability: persistent pleural effusion vs. entire liver transplant population. Survival at 2 years was 63.5% for persistent pleural effusion (PPEf) patients versus 79.1% for all orthotopic liver transplant (OLT) patients. When adjusted for age and Model for End-Stage Liver Disease (MELD) score, the hazard ratio for PPEf was 1.63 (95% CI 1.19–2.22, *p* = 0.002).

**Table 1 medsci-11-00024-t001:** Clinical characteristics and one-year mortality risk factors for persistent pleural effusion cohort.

	Total PPEf Patients (n = 124)	Alive at One Year (n = 97)	Deceased at One Year (n = 27)	*p*-Value
**Demographics**				
Age (years) ^a^	55.0 ± 9.3	54.1 ± 10.0	58.3 ± 5.5	0.04
Female ^b^	52 (41.9%)	43 (44.3%)	9 (33.3%)	0.38
MELD Score ^a^	28.2 ± 12.7	28.1 ± 12.8	28.4 ± 12.4	0.92
Charlson Comorbidity Index ^a^	5.9 ± 1.8	5.6 ± 1.7	6.8 ± 1.7	0.002
**Etiology of Liver Disease**				
Alcoholic Cirrhosis ^b^	38 (30.7%)	28 (28.9%)	10 (37.0%)	0.48
HBV Cirrhosis ^b^	13 (10.5%)	10 (10.3%)	3 (11.1%)	>0.99
HCV Cirrhosis ^b^	64 (51.6%)	49 (50.5%)	15 (55.6%)	0.67
NASH ^b^	61 (49.2%)	49 (50.5%)	12 (44.4%)	0.66
HCC ^b^	46 (37.1%)	34 (35.1%)	12 (44.4%)	0.38
**Pre-OLT Morbidity**				
Prior MI ^b^	6 (4.8%)	5 (5.2%)	1 (3.7%)	>0.99
COPD ^b^	5 (4.0%)	2 (2.1%)	3 (11.1%)	0.07
Asthma ^b^	7 (5.6%)	5 (5.2%)	2 (7.4%)	0.65
Portopulmonary Hypertension ^b^	7 (5.6%)	5 (5.2%)	2 (7.4%)	0.65
Hepatopulmonary Syndrome ^b^	29 (23.4%)	22 (22.7%)	7 (25.9%)	0.80
Hepatorenal Syndrome ^b^	71 (57.3%)	54 (55.7%)	17 (63.0%)	0.52
CKD ^b^	15 (12.1%)	8 (8.2%)	7 (25.9%)	0.02
Hemodialysis ^b,c^	56 (45.2%)	41 (42.3%)	15 (55.6%)	0.28
Ventilator dependence ^b,c^	39 (31.5%)	30 (30.9%)	9 (33.3%)	0.82
Vasopressor requirement ^b,c^	30 (24.2%)	24 (24.7%)	6 (22.2%)	>0.99
Admitted from home ^b^	45 (36.3%)	36 (37.1%)	9 (33.3%)	0.82
Thoracentesis ^b^	20 (16.1%)	17 (17.5%)	3 (11.1%)	0.56
Surgical Pleural intervention ^b^	4 (3.2%)	2 (2.1%)	2 (7.4%)	0.21
**Post-OLT Liver Function**				
Alanine transaminase ^a,d^	361 ± 493	354 ± 502	386 ± 468	0.77
Aspartate transaminase ^a,d^	497 ± 1067	409 ± 777.9	812 ± 1737	0.08
Total bilirubin ^a,d^	9.8 ± 9.7	8.2 ± 8.0	15.5 ± 13.0	<0.001
**Post-OLT Morbidity**				
Hemodialysis ^b,e^	53 (42.7%)	41 (42.3%)	12 (44.4%)	>0.99
Ventilator dependence ^b,e^	25 (20.2%)	20 (20.6%)	5 (18.5%)	>0.99
Vasopressor requirement ^b,e^	11 (8.9%)	8 (8.2%)	3 (11.1%)	0.70
LOS for index OLT hospitalization (days) ^f^	33 (19–75.5)	35 (20–77)	29 (16–60)	0.68
Total LOS in first year after OLT (days) ^f^	75.5 (50–129)	72 (47–117)	81 (64.5–156)	0.049
Readmissions in first year after OLT ^a^	3.1 ± 2.0	3.0 ± 1.9	3.4 ± 2.5	0.52
Total number of pleural interventions ^a^	2.27 ± 1.61	2.16 ± 1.49	2.66 ± 1.98	0.16
Surgical pleural intervention ^b,g^	53 (42.7%)	38 (39.2%)	15 (55.6%)	0.19
Positive PF Culture ^b^	14 (11.3%)	10 (10.3%)	4 (14.8%)	0.74

Abbreviations: CKD, chronic kidney disease; COPD, chronic obstructive pulmonary disease; HBV, hepatitis B virus; HCC, hepatocellular carcinoma; HCV, hepatitis C virus; LOS, length of stay; MELD, Model for End-Stage Liver Disease; MI, myocardial infarction; NASH, non-alcoholic steatohepatitis; OLT, orthotopic liver transplantation; PF, pleural fluid; PPEf, persistent pleural effusion. ^a^ = mean ± SD. ^b^ = n (%). ^c^ = within 48 h prior to transplantation. ^d^ = highest value on post-transplant days 8–365. ^e^ = for greater than 2 weeks after transplantation. ^f^ = median (IQR). ^g^ = tube thoracostomy, video-assisted thoracoscopic surgery (VATS) pleurodesis, and tunneled pleural catheter insertion.

**Table 2 medsci-11-00024-t002:** Clinical outcomes for post-OLT exudates by Light’s criteria (Exud^Light^) vs. transudates.

	Exud^Light^ (n = 111)	Transudate (n = 12)	Effect Size (95% CI)	*p*-Value
Age (years) ^a^	54.9 ± 9.4	56.4 ± 10.0		0.59
MELD Score ^a^	27.1 ± 12.7	34.6 ± 8.7		0.049
Charlson Comorbidity Index ^a^	5.9 ± 1.8	5.5 ± 1.3		0.46
**Outcomes**				
One-Year Survival ^b^	85 (78.0%)	10 (83.3%)	1.57 (0.36, 6.86) ^c^	0.55
Hemodialysis ^b,d^	44 (39.6%)	7 (58.3%)	0.83 (0.21, 3.27) ^e^	0.79
Ventilator dependence ^b,d^	22 (19.8%)	1 (8.3%)	3.13 (0.37, 26.14) ^e^	0.29
Vasopressor requirement ^b,d^	10 (9.0%)	0 (0.0%)	5.16 (0.24, 110.71) ^e,f^	0.29
LOS Post-Op (days) ^g^	33 (19–76)	29.5 (17.25–41)	0.4 (−0.09, 0.88) ^h^	0.11
Surgical Pleural Intervention ^b,i^	50 (45.0%)	3 (25.0%)	2.73 (0.68, 10.93) ^e^	0.16
Positive PF Culture ^b^	14 (12.6%)	0 (0%)	4.14 (0.21, 80.67) ^e,f^	0.35

Abbreviations: LOS Post-Op, length of stay post-operatively; MELD, Model for End-Stage Liver Disease; PF, pleural fluid. ^a^ = mean ± SD. ^b^ = n (%). ^c^ = hazard ratio. ^d^ = for greater than 2 weeks after transplantation. ^e^ = odds ratio. ^f^ = Firth bias correction used to generate odds ratio. ^g^ = median (IQR). ^h^ = mean difference. ^i^ = tube thoracostomy, video-assisted thoracoscopic surgery (VATS) pleurodesis, and tunneled pleural catheter insertion.

**Table 3 medsci-11-00024-t003:** Outcomes for exudates with elevated lactate dehydrogenase (Exud^LDH^) or with elevated protein (Exud^Prot^) vs. non-exudates.

	Exud^Prot^ (N = 75)	Non-Exud^Prot^ (N = 16)	Effect Size (95% CI)	*p*-Value	Exud^LDH^ (N = 63)	Non-Exud^LDH^ (N = 28)	Effect Size (95% CI)	*p*-Value
Age (years) ^a^	55.5 ± 8.5	54.6 ± 10.4		0.73	55.0 ± 8.6	56.0 ± 9.4		0.65
MELD Score ^a^	27.4 ± 12.8	31.9 ± 9.3		0.19	28.3 ± 11.7	27.8 ± 13.9		0.86
Charlson Comorbidity Index ^a^	6.0 ± 1.7	5.6 ± 1.9		0.40	5.9 ± 1.7	5.9 ± 1.6		0.99
**Outcomes** One-Year Survival ^b^	58 (79.5%)	13 (81.3%)	1.11 (0.32, 3.88) ^c^	0.87	51 (83.6%)	20 (71.4%)	0.56 (0.22, 1.42) ^c^	0.22
Hemodialysis ^b,d^	32 (42.7%)	7 (43.8%)	1.57 (0.43, 5.66) ^e^	0.49	28 (44.4%)	11 (39.3%)	1.44 (0.44, 4.66) ^e^	0.54
Ventilator Dependence ^b,d^	20 (26.7%)	1 (6.3%)	5.89 (0.72, 47.99) ^e^	0.10	19 (30.2%)	2 (7.1%)	5.59 (1.20, 26.04) ^e^	0.03
Vasopressor Requirement ^b,d^	10 (13.3%)	0 (0%)	7.55 (0.38, 148.44) ^e,f^	0.18	7 (11.1%)	3 (10.7%)	1.18 (0.26, 5.41) ^e^	0.83
LOS Post-Op (days) ^g^	36 (20–69)	30 (19–80)	0.17 (−0.26, 0.60) ^h^	0.42	48 (20.5–85)	29 (15–49)	0.38 (0.04, 0.73) ^h^	0.03
Surgical Pleural Intervention ^b,i^	30 (40.0%)	8 (50.0%)	0.69 (0.23, 2.07) ^e^	0.51	30 (47.6%)	8 (28.6%)	2.26 (0.87, 5.91) ^e^	0.10
Positive PF Culture ^b^	8 (10.7%)	2 (12.5%)	0.86 (0.16, 4.72) ^e^	0.87	9 (14.3%)	1 (3.6%)	4.48 (0.53, 37.72) ^e^	0.17

Abbreviations: LOS Post-Op, length of stay postoperatively; MELD, Model for End-Stage Liver Disease; PF, pleural fluid. ^a^ = mean ± SD. ^b^ = n (%). ^c^ = hazard ratio. ^d^ = for greater than 2 weeks after transplantation. ^e^ = odds ratio. ^f^ = Firth bias correction used to generate odds ratio. ^g^ = median (IQR). ^h^ = mean difference. ^i^ = tube thoracostomy, video-assisted thoracoscopic surgery (VATS) pleurodesis, and tunneled pleural catheter insertion.

**Table 4 medsci-11-00024-t004:** Outcomes for pleural effusions with neutrophil (Ne) predominance Or lymphocyte (Lym) predominance vs. no predominance.

	Ne-Predominance (N = 26)	No Ne- Predominance (N = 86)	Effect Size (95% CI)	*p*-Value	Lym- Predominance (N = 33)	No Lym- Predominance (N = 79)	Effect Size (95% CI)	*p*-Value
Age (years) ^a^	55.7 ± 6.6	54.3 ± 10.3		0.52	53.8 ± 10.5	54.9 ± 9.2		0.55
MELD Score ^a^	26.5 ± 12.0	27.5 ± 13.0		0.74	26.2 ± 12.9	27.7 ± 12.7		0.59
Charlson Comorbidity Index ^a^	6.2 ± 1.5	5.8 ± 1.9		0.40	5.9 ± 1.9	5.9 ± 1.8		0.96
**Outcomes** One-Year Survival ^b^	17 (65.4%)	70 (81.4%)	2.16 (0.9, 5.16) ^c^	0.08	26 (78.8%)	61 (77.2%)	1.14 (0.46, 2.81) ^c^	0.77
Hemodialysis ^b,d^	10 (38.5%)	35 (40.7%)	1.03 (0.33, 3.18) ^e^	0.96	11 (33.3%)	34 (43.0%)	0.72 (0.24, 2.19) ^e^	0.57
Ventilator Dependence ^b,d^	9 (34.6%)	13 (15.1%)	3.10 (1.12, 8.55) ^e^	0.03	2 (6.1%)	20 (25.3%)	0.19 (0.04, 0.88) ^e^	0.03
Vasopressor Requirement ^b,d^	4 (15.4%)	4 (4.7%)	10.96 (1.46, 82.15) ^e^	0.02	0 (0%)	8 (10%)	0.10 (0.01, 1.81) ^e,f^	0.12
LOS Post-Op (days) ^g^	35.5 (17–56.5)	30.5 (19–74)	−0.04 (−0.4, 0.32) ^h^	0.82	25 (19–48)	36 (18.5–78.5)	−0.28 (−0.6, 0.05) ^h^	0.09
Surgical Pleural Intervention ^b,i^	15 (57.7%)	28 (32.6%)	2.88 (1.17, 7.10) ^e^	0.02	11 (33.3%)	32 (40.5%)	0.72 (0.31, 1.70) ^e^	0.45
Positive PF Culture ^b^	5 (19.2%)	8 (9.3%)	2.56 (0.74, 8.91) ^e^	0.14	0 (0%)	13 (16.5%)	0.07 (0.01, 1.17) ^e^	0.06

Abbreviations: LOS Post-Op, length of stay postoperatively; MELD, Model for End-Stage Liver Disease; PF, pleural fluid. ^a^ = mean ± SD. ^b^ = n (%). ^c^ = hazard ratio. ^d^ = for greater than 2 weeks after transplantation. ^e^ = odds ratio. ^f^ = Firth bias correction used to generate odds ratio. ^g^ = median (IQR). ^h^ = mean difference. ^i^ = tube thoracostomy, video-assisted thoracoscopic surgery (VATS) pleurodesis, and tunneled pleural catheter insertion.

## Data Availability

The data presented in this study are available upon request from the corresponding author.

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
