# Peer review of "Implications of Pleural Fluid Composition in Persistent Pleural Effusion following Orthotopic Liver Transplant"

_medsci, 2023, doi:10.3390/medsci11010024_

Round 1
Reviewer 1 Report
The manuscript is easy to read, well-structured and covering a very interesting topic from a new perspective. It encourages further research concerning causes of post-transplant pleural effusions, and so potentially improving patients’ outcomes by better (or sooner implemented) therapies.
The conclusions are consistent with the argumentation; the subject is interesting, and only few publications are covering it.
Author Response
Dear Reviewer 1,
We thank you for your positive feedback on our manuscript. We are grateful to be under consideration for publication in Medical Sciences.
On behalf of the co-authors, sincerely,
Bhavesh H. Patel
M.D. Candidate, Class of 2024
David Geffen School of Medicine at UCLA
Email: bhaveshpatel@mednet.ucla.edu
Reviewer 2 Report
The authors presented a retrospective cohort study that evaluated the clinical, biochemical, and cellular characteristics of post-orthotopic liver transplant patients with persistent pleural effusions. Interestingly, they found that using LDH levels and incorporating the cellular analysis was useful in predicting posttransplant morbidity, which could contribute to the better management of orthotopic liver transplant patients.
Author Response
Dear Reviewer 2,
We thank you for your positive feedback on our manuscript. We are grateful to be under consideration for publication in Medical Sciences.
On behalf of the co-authors, sincerely,
Bhavesh H. Patel
M.D. Candidate, Class of 2024
David Geffen School of Medicine at UCLA
Email: bhaveshpatel@mednet.ucla.edu
Reviewer 3 Report
Patel and colleagues conducted a retrospective analysis on the characteristics and outcomes of persistent pleural effusions following orthotopic liver transplantation in adult patients. They classified pleural effusions as exudative or transudative, as well neutrophilic and lymphocytic and determined that exudative and neutrophil-predominant persistent effusions were associated with worse clinical outcomes.
I would like to raise the following major issues regarding the methodological accuracy and integrity of this study: 1) The authors define persistent pleural effusion using a definition of pleural effusion present for at least 30 days, however, compare short-term postoperative outcomes (e.g., vasopressor use, ventilator dependence,). According to the literature, the median LOS following OLT is 14 days (1). The authors in this study, provide an average LOS of 54 days very large SDs. The authors do not provide an explanation for this large difference in LOS. The presence of outliers likely has a significant impact on the mean and thus the median (range) or median (1st IQR - 3rd IQR) may be a more appropriate measure for this outcome. 2) Given the above data, it is likely that many of their patients were discharged prior to the 30-day period. Thus, these short-term outcomes appear to be out of context as patients with these complications require care in an ICU setting. In addition, they most likely occured before a persistent pleural effusion was determined to be present by definition. 3) The authors fail to provide data for significant outcomes that would be more relevant to patients with a complication that occurs by definition at least 30 days post-transplant. For example, number of interventions such as thoracenteses required; incidence of empyema; readmission rates etc. 4) Odds ratios were not calculated for some important outcomes, due to zero events noted in one of the groups. For example, positive pleural fluid culture in the ExudLight/Non-ExudLight groups. and vasopressor requirement in the ExudProt/Non-ExudProt groups. There are widely accepted statistical methods (e.g., the Haldane-Ascombe correction) which can be used to overcome this issue. 5) The authors argue that Light's criteria lack specificity for identifying exudates and employ two different one-criterion (protein or LDH) methods to make a distinction between the two, which would expectedly increase specificity but decrease the sensitivity of identifying an exudate. Their results show that exudates are associated with worse clinical outcomes, which would logically imply that a high sensitivity would be more clinically useful in this case. The difference in the sample sizes in the exudate and transudate groups as a result of the sensitivity/specificity trade-off of each method results in a much smaller sample size for the transudate group (n=12) compared to using the LDH (n = 28) or protein criterion only (n = 16). It is likely that many of the comparisons using the Light criteria result in non-statistically significant differences as a result of sample size restrictions; notable differences exist in many of the outcomes between the ExudLight and Non-ExudLight groups i.e., positive fluid culture, ventilator dependence that are likely not statically significant as a result of inadequate power to detect these differences rather than the Light criteria being less sensitive in detecting these differences. In fact, the Light criteria have already been shown to be more accurate in distinguishing exudates from transudates more accurately than any single pleural fluid test (2). 6) In their survival analysis (Figure 1), the proportionality-of-hazards may be violated as the two curves clearly cross each other. Thus, the use of a parametric time-to-event estimate (hazard ratio) may be inappropriate. The authors should consider using an appropriate statistical test to ensure the proportionality-of-hazards assumption is true (e.g., Grambsch–Therneau test for a non-zero slope, scaled Schoenfeld residuals, log–log survival plots, predicted versus observed survival function) and use non-parametric effect estimates (such as the restricted mean survival time, life expectancy difference, or life expectancy ratio) in case the assumption is violated. 7) The authors use odds ratios and arbitrary single time points (e.g., 1-year survival), to compare survival outcomes, where in fact survival is a time-to-event outcome. Thus, the use of odds-ratios is an inappropriate outcome measure, omitting important information from the survival curve. Therefore, a survival analysis using outcome measures (eg. hazard ratios) would be needed. As a result of the multiple methodological shortcomings of this study and the resulting lack of applicability to clinical practice, I am unfortunately unable to recommend this article for publication. References:1) Amiri M, Toosi MN, Moazzami B, et al. Factors Associated With Length of Hospital Stay Following Liver Transplant Surgery. Exp Clin Transplant. 2020;18(3):313-319. doi:10.6002/ect.2019.0077
2) Porcel JM, Light RW. Pleural Fluid Analysis: Are Light's Criteria Still Relevant After Half a Century?. Clin Chest Med. 2021;42(4):599-609. doi:10.1016/j.ccm.2021.07.003Author Response
Please see the attachment.

Round 2
Reviewer 3 Report
I appreciate all the authors for taking the time to provide detailed responses to all my queries, using appropriate references to support their answers. I would also like to thank the authors for making significant additions and revisions to the methodology and content of their original manuscript.
Following their revisions, I believe that the significance and limitations of their findings are now presented in a more clear way to the readers.
Thus, I recommend that the manuscript can be accepted for publication in its present form.